# TANGO: TOKENIZED ANALYSIS-BY-SYNTHESIS FOR 3D NUCLEI SEGMENTATION CORRECTION

## ABSTRACT

Accurate 3D nuclei segmentation underpins studies from development and regeneration to large-scale anatomy. Landmark volumetric EM datasets, whole-brain Drosophila (FAFB) and petabyte-scale mouse brain tissue (MICrONS), have enabled cellular-scale mapping, yet their released nuclei segmentations retain errors despite extensive proofreading. The *last mile*—rare, heterogeneous false merges/splits and missing-slice or misalignment artifacts—remains difficult, where discriminative correction models overfit to training degradations and generalize poorly. We introduce **TANGO**, a tokenized *analysis-by-synthesis* framework for 3D nuclei segmentation correction. TANGO tokenizes the erroneous seed mask into sub-nucleus fragments and *generatively* completes multiple shape hypotheses conditioned on the image and tokens. Training applies *slice-patch masking* to complete-nucleus annotations (without using error labels). A lightweight *ordinal* selector ranks overlapping hypotheses, and simple NMS decodes a reliable subset of fixes. To evaluate at the brain scale, we curate **NucEMFix**, a systematic benchmark of nuclei *error cases* across FAFB and MICrONS (8,000 + annotated error nuclei). Beyond EM, we assess generality on public *C. elegans* L1 *confocal* volumes. TANGO consistently improves F1 over strong baselines, achieving state-of-the-art correction without prompt engineering or error-specific supervision. We release NucEMFix, code, and evaluation scripts for reproducible assessment and for quantifying proofreading-time savings.

## 1 INTRODUCTION

Accurate 3D nuclei segmentation provide the foundation for a wide range of biological studies—from development and regeneration to large-scale anatomy (Caicedo et al., 2019; Nunley et al., 2024). In neuroscience, nuclei provide spatial anchors that link cellular identity to circuit reconstruction and cross-scale integration (Jorstad et al., 2016; Schneider-Mizell et al., 2025; Kuan et al., 2020; Lee et al., 2025), and segmentation quality directly impacts downstream analyses (Pang et al., 2025; Consortium et al., 2021; Chen et al., 2025; Wang et al., 2024a). As shown in Fig. 1, landmark EM datasets such as FAFB and MICrONS have unlocked cellular-scale mapping at unprecedented scope, yet their released nuclei segmentations still contain remaining error despite extensive proofreading. The remaining *last mile* is dominated by rare, heterogeneous failures—false merges/splits and slice-missing or misalignment artifacts—on which traditional discriminative correction models, tuned to specific degradations, generalize poorly and often overfit (Aswath et al., 2023; Plaza, 2014).

Most prior 3D correction pipelines in connectomics operate by oversegmentation and pair-matching (Chen et al., 2024; Berman et al., 2022; Matejek et al., 2019) . These approaches face two structural issues: (i) they rely on paired error labels, which are scarce, variable across datasets, and expensive to obtain at the brain scale; and (ii) they struggle to *complete* missing structure (e.g., through axial gaps or misalignments) and to split large false merges without additional human input.

We introduce **TANGO** (**T**okenized **A**nalysis-by-Synthesis for 3D **N**uclei Se**G**mentation C**O**rrection), a tokenized framework that treats correction as *analysis by synthesis*. TANGO first tokenizes the erroneous seed mask into sub-nucleus fragments, then *generatively* completes multiple shape hypotheses conditioned on the image and tokens. Training uses *slice-patch masking* learned solely from complete nuclei—no error labels—so the model learns a completion prior that is robust to anisotropy,

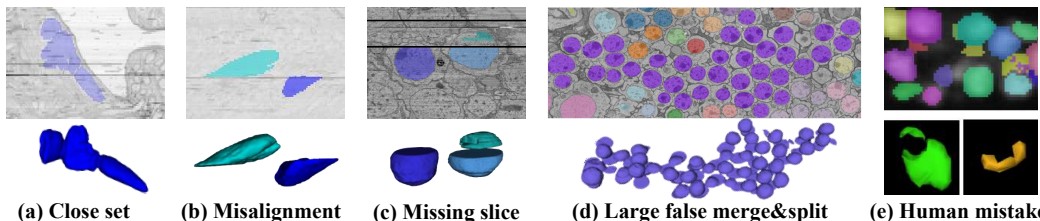

**(a) Close set**  **(b) Misalignment**  **(c) Missing slice**  **(d) Large false merge&split**  **(e) Human mistakes**

Figure 1: The long-tail of nuclei annotation errors in public 3D microscopy datasets. Representative modes: (a) close set contact with ambiguous boundaries yielding false merges; (b) misalignment with discontinuous objects; (c) missing slices with axial gaps with no image evidence; (d) large merge–split hybrids due to misalignment and high packing density; (e) annotation-induced artifacts.

slice dropouts, and local artifacts. In inference, a lightweight *ordinal* selector ranks overlapping hypotheses, and a simple NMS decodes a reliable subset of fixes.

To evaluate correction at the brain scale, we curate **NucEMFix**, a systematic, brain-wide benchmark of nuclei *error cases* across FAFB and MICrONS, comprising 8,000 + annotated error nuclei spanning false merges, false splits, and false merges&splits. Beyond EM, we assess generality on public *C. elegans* L1 *confocal* volumes (Weigert et al., 2020).

Our paper has the following contributions:

- **TANGO:** a tokenized *analysis-by-synthesis* framework that learns completion from *complete nuclei only* via *slice-patch masking*, and selects overlapping hypotheses with an ordinal (pairwise) head and NMS.
- **NucEMFix:** a brain-wide benchmark of nuclei error cases in large-scale EM volumes (FAFB, MICrONS) with diverse, realistic failures to drive research on last-mile correction.
- **State-of-the-art performance:** Reproducible gains on EM and confocal datasets over modern baselines, without prompts or corrected labels.

## 2 RELATED WORKS

**3D nuclei instance segmentation datasets** Public benchmarks have driven progress across modalities and species, including *C. elegans* (Long et al., 2009), *Parhyale* (Alwes et al., 2016), rodent and human tissue (Tokuoka et al., 2020; Caicedo et al., 2019; Ruszczycki et al., 2019), and cancer cell lines (Ulman et al., 2017). More recently, NucMM (Lin et al., 2021) provided large-scale microCT & EM nuclei volumes, and NIS3D (Zheng et al., 2023) released densely annotated 3D fluorescence data. These resources focus on *segmenting* all nuclei in a volume. Our **NucEMFix** complements them by targeting the *last-mile error space* in large-scale EM (false merges/splits, slice-missing/misalignment), supplying brain-wide, curated *error cases* for quantitative *correction* benchmarking. Unlike prior datasets, NucEMFix is coupled with training strategies that require *only complete nuclei* (no error labels), aligning with TANGO's design.

**Promptable and foundation-model segmentation** Promptable segmenters (e.g., SAM (Kirillov et al., 2023) and SAM-Med3D (Wang et al., 2024b)) produce instance masks conditioned on points, boxes, or masks, and excel for interactive delineation and broad generalization. However, last-mile nuclei repair poses different demands: (i) no prompts at the brain scale; (ii) *completion* through anisotropic slice gaps; and (iii) splitting large false merges into multiple instances. Even when adapted as "correctors" with mask prompts, such models mainly refine boundaries and are sensitive to prompt design and absolute quality calibration. In contrast, TANGO generates multiple completion hypotheses from a tokenized seed and uses *ordinal* (pairwise) selection to robustly choose among overlaps.

**Automatic error correction for instance segmentation** In connectomics, automatic detection/correction has been explored via multiscale CNNs that classify split/merge errors (Zung et al., 2017), graph/oversegmentation with pair matching (Matejek et al., 2019; Berman et al., 2022; Chen et al., 2024), and workflow accelerators that rely on labeled error pairs or carefully tuned heuristics.

Figure 2: TANGO pipeline. (a) **Tokenization:** realign slices and apply a DT–watershed to split the erroneous seed $M^0$ into sub-nucleus tokens. (b) **Token completion:** a 3D U-Net (Çiçek et al., 2016) takes each token with its local crop and proposes mask hypotheses with quality scores. (c) **Ordinal selection & decoding:** rank hypotheses with an Ordinal Quality Estimator and keep a reliable set via NMS.

These approaches face two limitations at the brain scale: (i) paired error annotations are scarce and dataset-specific; (ii) they struggle to *reconstruct missing structure* (e.g., slice dropouts, misalignments) and to resolve very large merges without human input. We formulate nuclei correction as *completion* rather than pair matching: TANGO learns from complete nuclei using *slice-patch masking* to simulate realistic failures, enumerates token-conditioned hypotheses, and selects them ordinally before a simple NMS. This yields preserved, topology-consistent fixes that generalize across modalities without prompts or error-specific supervision.

## 3 METHOD

### 3.1 TANGO OVERVIEW

Let $I$ be a 3D volume with an initial instance mask $M^0$. The goal is to produce corrected instances $\hat{S} = \{\hat{M}_k\}_{k=1}^K$ for last-mile failures (false merges, false splits, and false merge&splits). Unlike discriminative predictors that map $(I, M^0)$ to a single mask and overfit to seen degradations, TANGO follows an *analysis-by-synthesis* strategy: generate token completions under a learned shape prior, then *select* reliably using ordinal evidence rather than absolute calibration.

Figure 2 summarizes the three modules of TANGO:

1. **Tokenization**: decompose $M^0$ into *sub-nucleus tokens* $\mathcal{T} = T(M^0) = \{t_j\}_{j=1}^J$ via realignment and DT–watershed; Each token is a compact local fragment.

2. **Token completion**: for each $t_j$, the network $G_\theta$ proposes hypotheses $\{M_j\}$ from $(I, t_j)$. $G_\theta$ is trained only on *complete* nuclei using synthetic data generated by *slice-patch masking*, learning a completion prior without error labels.

3. **Ordinal selection & decoding**: an Ordinal Quality Estimator $S_\phi$ ranks hypotheses; NMS retains a reliable subset $\hat{S} \subseteq \{M_j\}$.

### 3.2 TOKENIZATION

Tokenization splits the erroneous seed $M^0$ into *sub-nucleus tokens* via slice realignment followed by a DT–watershed. Tokens isolate convex lobes and serve as compact completion prompts, promoting diverse hypotheses.

**Domain-specific Pre-processing**. Microscopy modalities exhibit distinct degradations that can impact segmentation. For example, local slice misalignments are common for large volumetric EM, so we realign the local crop of each instance. Given a volume $I \in \mathbb{R}^{Z \times H \times W}$, we estimate per-slice in-plane shifts $D = \{(D_z^x, D_z^y)\}_{z=1}^Z$ via patch-based phase correlation to the nearest valid neighbor. Slices with excessive background or abnormally low intensity variance are marked invalid, assigned zero shift, and their masks are discarded to prevent error propagation. We accumulate offsets to form

Figure 3: Token completion module. (a) **Training Data Synthesis :** We synthesize training data by applying *slice-patch masking* to both the mask and its cropped image, producing corrupted inputs for the Completion Network. (b) **Completion Network** $G_\theta$ **and Ordinal Quality Estimator (OQE)** $S_\phi$ **Training:** $G_\theta$, a 3D U-Net, learns to complete corrupted tokens with completion loss $\mathcal{L}C$, while $S\phi$, consisting of global average pooling (GAP) and an MLP, learns to preserve hypothesis ordering with ranking loss $\mathcal{L}_R$; the two are trained jointly. Both objectives are jointly optimized.

$D$, warp each crop with $D$, and map predictions back using the inverse field. Details are provided in A.7.

**Sub-nucleus Token Generation**. We first apply Gaussian smoothing to the image $I$ to reduce noise and suppress spurious peaks, and then compute the gradient magnitude map $G$ from $I$. For each mask, we compute a distance transform, extract local maxima with peak radius $r$ which is default set to 10, and use them to seed a watershed on the gradient map $G$. The resulting fragments $\{t_j\}_{j=1}^J$ are the tokens. Very small fragments ($|t_j| < v_{\min}$) are merged into neighbors; the remaining tokens are passed to completion.

## 3.3 TOKEN COMPLETION

As shown in Fig. 3 the token completion module consists of two steps: (1) training data synthesis, (2) joint training of the completion network $G_\theta$ and the Ordinal Quality Estimator $S_\phi$.

**Objective.** Given $K$ tokens $\{t_j\}_{j=1}^K$ with corresponding local crops $\{(I_j, M_j^0)\}_{j=1}^K$, the model predicts, for each token $t_j$, a completed instance and its quality score:

$$(\hat{M}_j, \hat{q}_j) = \Big( G_\theta(X_j), \ S_\phi\big(X_j, G_\theta(X_j)\big)\Big), \quad X_j = [I_j; t_j], \ j = 1, \dots, K.$$

Here, $G_\theta$ is a 3D U-Net completion network that predicts completed instances, $S_\phi$ is the ordinal quality estimator that assigns a quality score $\hat{q}_j$ to each hypothesis, and $X_j$ stacks the image crop and token mask as input channels.

**Training Data Synthesis**. We adopt a permissive strategy for sampling complete nuclei: bounding-box thresholds are used to exclude potential false merges, while voxel-count thresholds filter out potential false splits. From complete nuclei $(I, M)$ we synthesize coupled corruptions $(\tilde{I}, \tilde{M}) = \mathcal{C}(I, M)$ using (i) *slice* masking (runs of mask slices with only a few adjacent image slices occluded) and (ii) *patch* masking (foreground mask blocks paired with subsampled image blocks from the same region) (Fig. 3 (a)). During training, $\tilde{M}$ is embedded into a context crop with distractor fragments to mimic real seeds. The network learns a conditional inverse by mapping $(\tilde{I}, \tilde{M}, t_j)$ to $M$, with completion loss:

$$\mathcal{L}_C = \mathcal{L}_{\text{Dice}}(\hat{M}, M).$$

In parallel, Ordinal Quality Estimator $S_\phi$ is trained with a pairwise ranking loss $\mathcal{L}_R$ to order hypotheses (Fig. 3 (b)).

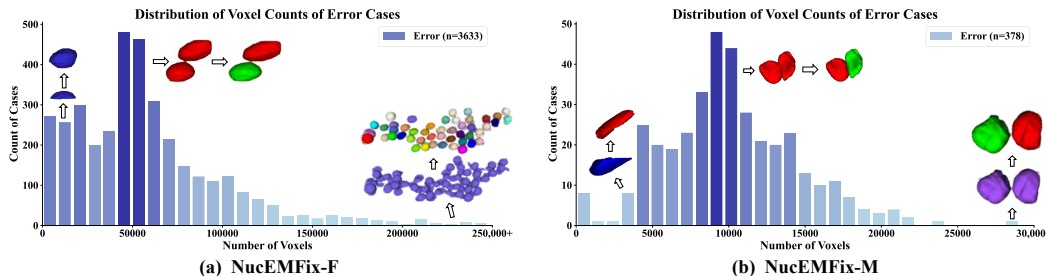

(a) NucEMFix-F                                    (b) NucEMFix-M

Figure 4: Distribution of error case sizes in the proposed NucEMFix dataset, showing a long-tail distribution. Error cases are distributed throughout the whole brain and vary widely in shape and size, categorized into three types: false merge, false split, and false merge&split.

### 3.4 ORDINAL SELECTION AND LOSSRANK-NMS

Token completion produces overlapping hypotheses; we select a subset.

**Ordinal Quality Estimator (OQE)**. Instead of regressing an absolutely calibrated IoU, we train $S_\phi$ to preserve the *ordering* of hypothesis quality. As illustrate in Fig. 3 (b), $S_\phi$ consists of a global average pooling layer followed by a multi-layer perceptron (MLP). It takes the feature map from the last layer of the 3D U-Net and predicts the quality score of each hypothesis. For a minibatch, we sample disjoint pairs $(i, j)$ and use a pairwise ranking loss:

$$\mathcal{L}_{\mathrm{R}}^{(i,j)} = \max\big(0, \ -s_{ij}\,[\,\hat{l}_i - \hat{l}_j\,] + \xi\big), \quad s_{ij} = \begin{cases} +1, & l_i > l_j \\ -1, & \text{otherwise} \end{cases}, \tag{1}$$

where $\hat{l}_i = S_\phi(M_i)$ represents the quality score assigned by the OQE to hypothesis $M_i$, and $\xi$ is a small margin which is default set to 0.05. This ordinal formulation is more stable under distribution shift than absolute regression.

**Decoding with Non-maximum Suppression (NMS)**. At inference, we score $M$ by $q(M) = -\hat{l}(M)$, sort hypotheses by $q$, and greedily apply NMS with an overlap metric derived from IoU:

$$r(A \to B) = \frac{|A \cap B|}{|A|}, \qquad \text{suppression if } \max\{r(A \to B), r(B \to A)\} > \tau.$$

This metric is particularly effective when multiple hypotheses correspond to the same nucleus but differ in size. It encourages suppression of smaller or lower-quality hypotheses even when conventional IoU would remain high. The union over tokens yields the corrected set $\hat{\mathcal{S}}$.

### 3.5 LEARNING AND INFERENCE

The joint objective is

$$\mathcal{L} = \frac{1}{B} \sum_i \mathcal{L}_{\mathrm{C}}(\hat{M}_i, M^i) + \frac{\lambda}{B/2} \sum_{(i,j)} \mathcal{L}_{\mathrm{R}}^{(i,j)}, \tag{2}$$

where $\lambda$ is a balanced weight that is default set to 1.

**Learning**. We jointly train $G_\theta$ and $S_\phi$ with Eq. 2. A 5-epoch warm-up optimizes only $\mathcal{L}_{\mathrm{C}}$; afterward we optimize $\mathcal{L}_{\mathrm{C}} + \mathcal{L}_{\mathrm{R}}$. To synthesize realistic degradations, we apply *slice-patch masking* where image masking is a strict subset of segmentation masking, with spans/ratios uniformly sampled. This keeps corruption local and aligned, encouraging $G_\theta$ to segment visible regions and complete missing structure via learned shape priors.

**Inference**. For each token $t_j$ from the realigned seed $(I_r, M_{0,r})$: (i) crop a cubic patch at its centroid, (ii) run $G_\theta$ to produce hypotheses $\{M_{j,k}\}$ and score them with $S_\phi$, (iii) apply NMS within the crop, and (iv) map survivors back via the inverse realignment. The union over tokens yields the corrected set $\hat{\mathcal{S}}$.

Table 1: **NucEMFix statistics.** "Initial nuclei" counts the released instances before curation. "Error-affected nuclei" count nuclei impacted by each error type.

| Subset | Volume size (vox) | Initial nuclei | Error-affected nuclei | | | |
|---|---|---|---|---|---|---|
| | | | False merge | False split | False merge&split | Total |
| NucEMFix-F | $3584 \times 8192 \times 14336$ | 106,978 | 5,258 | 1,031 | 1,012 | 7,301 |
| NucEMFix-M | $1632 \times 4096 \times 6144$ | 77,475 | 711 | 20 | 26 | 757 |

## 4 NUCEMFIX DATASET

**Dataset Source**. NucEMFix is curated from two large public EM datasets: the Full Adult Fly Brain (FAFB) (Zheng et al., 2018) and MICrONS ($1 \, mm^3$ *Mouse* visual cortex) (Consortium et al., 2021). For FAFB, we start from the released whole-brain nuclei instances (Mu et al., 2021); for MICrONS, we crop a central subvolume of the released nuclei instances to avoid boundary artifacts.

**Dataset Annotation**. (1) To prioritize review, we apply multi-step morphological erosion and flag samples that split under erosion (merge candidates), and rank by voxel count to surface size extremes (large = merge, small = split). (2) All nuclei are then inspected in Neuroglancer (Maitin-Shepard, 2016) to label false merges, false splits, and false merge&split (hybrid) cases, as well as misalignment artifacts. (3) Errors are corrected in ITK-SNAP (Yushkevich et al., 2016) by three trained annotators (~250 hours). Two experts subsequently double-review all edits to consensus.

**Dataset Statistics**. Across both subsets, errors are long-tailed: a few very large merge cases dominate, while many small split fragments populate the tail (Fig. 4). NucEMFix-F also contains larger nuclei on average than NucEMFix-M, leading to more extensive error regions. As shown in Tab. 1, NucEMFix-F consists of $3584 \times 8192 \times 14336$ voxels at $80 \times 64 \times 64 \, nm^3$ resolution, containing $106,978$ neuronal nuclei, among which we annotated $>3,000$ error cases affecting $>7,300$ ($\sim 7\%$) nuclei. NucEMFix-M comprises $1632 \times 4096 \times 6144$ voxels at the same resolution, with $77,475$ nuclei and $\sim 400$ error cases affecting $>750$ nuclei. Note that the counts above denote *nuclei affected*, since a single error case may involve multiple nuclei, especially in false merge cases. The mouse brain is substantially larger than that of Drosophila, resulting in more sparsely distributed nuclei and consequently fewer segmentation errors compared to FAFB. In MICrONS, we observed that most errors occur in the vicinity of blood vessels, affecting endothelial and perivascular cell nuclei. These nuclei often exhibit irregular morphologies, leading to false merges or fragmented instances, and thus represent a key target for future correction strategies.

## 5 EXPERIMENTS ON NUCEMFIX

### 5.1 EXPERIMENT SETTINGS

**Evaluation Metrics**. To evaluate performance, we adapt standard 3D instance segmentation metrics, precision, recall, and F1 score. A predicted nucleus and a ground-truth instance are considered a match if their Intersection-over-Union (IoU) is at least $\tau$. Unless otherwise noted, we use $\tau = 0.75$ (Weigert et al., 2020).

**Baseline Comparisons**. We compared our approach against pair-matching-based error correction methods in connectomics, with each baseline first processed using our tokenization procedure. We evaluated two naive baselines, *(1) Nearest Neighbor*: two sub-nucleus tokens are merged if their centroids are within a distance of $d = 20$ voxels and the angle between their centroid-connecting line and the $xy$-plane normal is $\leq 30°$, *(2) Ellipsoid Fitting*: Considering the ellipsoidal shape of nuclei, each sub-nucleus token is fitted with a 3D ellipsoid using linear least squares (LLSQ), and two tokens are merged if the proportion of intersecting surface points between their fitted ellipsoids exceeds a threshold $\tau = 0.3$. As well as two learning-based approaches: *(3) Berman et al. (2022)*: which classifies token pairs using PointNet++ (Qi et al., 2017) on sampled point clouds, *(4) Chen et al. (2024)*: which combines contrastive embeddings with shape features for pairwise classification using PointNet++. In contrast, *(5) SAM-Med3D* (Wang et al., 2024b) follows a completion-based paradigm, using each token as a prompt to complete a full nucleus.

## 5.2 MAIN RESULTS

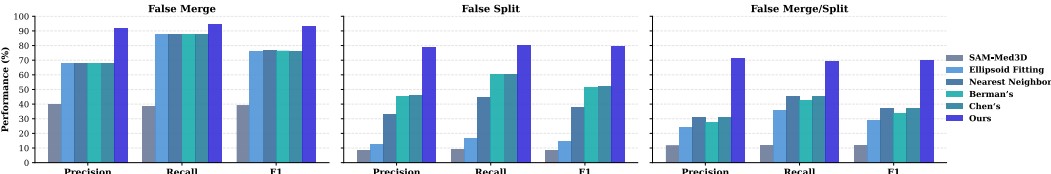

Figure 5: Performance comparison of different methods on the three error types in the NucEMFix-F

**Quantitative Results**. Tab. 2 reports the benchmark results on the proposed NucEMFix dataset. Among the two naive baselines, the ellipsoid fitting approach underperforms the nearest-neighbor method, particularly on FAFB, though both achieve relatively better results on MICrONS where false split cases are rare. Chen's method shows a pronounced performance gap between datasets, as the densely packed and small nuclei in MICrONS induce frequent erroneous merges when local features are projected to point clouds. Even with fine-tuning, SAM-Med3D fails to reliably complete nuclei (Fig. 6 (g)) due to the substantial domain gap and the scale differences between EM and medical imaging data. By contrast, our SCNF consistently surpasses all competing methods across metrics. On MICrONS, the gains of our approach over baselines are smaller than on FAFB. This is largely because false merges dominate in MICrONS (Tab. 1), and our tokenization applied to all baselines—already mitigates most of these errors. Furthermore, severe misalignment in this dataset hampers the resolution of false split errors, posing a challenge that limits the performance of all methods.

**Qualitative Results**. The error correction visualization on NucEMFix-F is shown in Fig. 6. For the pair matching based approach, the input consists of the error segmentation to be corrected and its neighboring segmentation candidates. As shown, only Berman's method achieves partial success, while others fail to establish effective correspondences. However, Berman's method struggles with merging the intermediate sheet-like instances. The nuclei completed by the fine-tuned SAM-Med3D exhibit highly irregular shapes. In contrast, our method produces a more accurate and complete correction.

## 5.3 ABLATION STUDIES

To assess the contributions of TANGO's core components, we perform comprehensive ablation studies on the NucEMFix-F dataset, evaluating the effects of (i) Training Data Synthesis, (ii) the Ordinal Quality Estimator (OQE), and (iii) the feature map used for hypothesis quality prediction. Additional ablation studies on realignment pre-processing and hyperparameter settings are provided in Section A.4.

**Training Data Synthesis**. We ablate different corruption strategies by evaluating variants without slice masking, patch masking, or any image masking (Tab. 3). The model performs poorly without slice masking, especially on false splits, because EM volumes are $z$-stacked and missing slices—common sources of errors—cannot be adequately addressed by patch masking alone. The *slice-patch masking* performs best: slice masking enables recovery from axial gaps by generating

Table 2: Benchmark results on the proposed NucEMFix dataset.

| Method | Train data | NucEMFix-F | | | NucEMFix-M | | |
|---|---|---|---|---|---|---|---|
| | | Pre/% | Rec/% | F1/% | Pre/% | Rec/% | F1/% |
| Nearest Neighbor | N/A | 57.44 | 75.93 | 65.40 | 66.53 | 65.03 | 65.77 |
| Ellipsoid Fitting | N/A | 53.43 | 70.58 | 60.82 | 66.71 | 64.77 | 65.73 |
| Berman et al. (2022) | Synthesis | 58.28 | 77.73 | 66.62 | 65.04 | 63.58 | 64.30 |
| Chen et al. (2024) | Synthesis | 58.99 | 77.93 | 67.15 | 49.86 | 48.87 | 49.36 |
| SAM-Med3D | Synthesis | 47.18 | 48.67 | 47.91 | 27.40 | 23.84 | 25.50 |
| **Ours** | Synthesis | **87.17** | **89.26** | **88.20** | **70.18** | **67.02** | **68.56** |

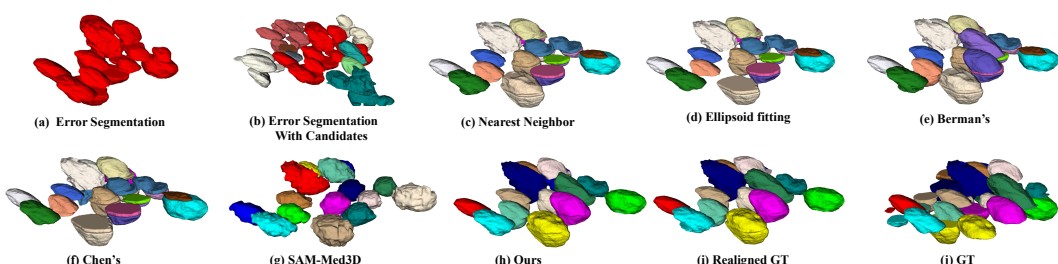

(a) Error Segmentation  (b) Error Segmentation With Candidates  (c) Nearest Neighbor  (d) Ellipsoid fitting  (e) Berman's

(f) Chen's  (g) SAM-Med3D  (h) Ours  (i) Realigned GT  (j) GT

Figure 6: Qualitative nuclei error correction results on the NucEMFix-F. Different colors indicate different instances. The pair-matching method achieves a low success rate, and Berman's approach struggles to merge sheet-like instances. Fine-tuned SAM-Med3D lacks precise completion of the nuclei. In contrast, our method delivers the best performance.

Table 3: Ablation studies on different corruption strategies in training. *slice-patch masking*, by combining slice, patch, and image masking, enables comprehensive modeling of missing structures, leading to superior segmentation completion performance.

| Setting | False Merge | | | False Split | | | False Merge&Split | | | All | | |
|---|---|---|---|---|---|---|---|---|---|---|---|---|
| | Pre/% | Rec/% | F1/% | Pre/% | Rec/% | F1/% | Pre/% | Rec/% | F1/% | Pre/% | Rec/% | F1/% |
| w/o slice masking | 72.86 | 87.72 | 79.60 | 14.65 | 17.66 | 16.02 | 27.31 | 35.48 | 30.87 | 57.94 | 70.54 | 63.62 |
| w/o patch masking | 90.27 | 94.02 | 92.21 | 75.63 | 78.93 | 77.24 | 68.20 | 67.94 | 68.07 | 85.22 | 88.24 | 86.71 |
| w/o image masking | 90.76 | 94.23 | 92.46 | 77.79 | 79.90 | 78.83 | 69.28 | 68.33 | 68.80 | 86.05 | 88.59 | 87.30 |
| slice-patch masking | **91.73** | **94.82** | **93.25** | **78.67** | **80.58** | **79.62** | **71.29** | **69.40** | **70.33** | **87.17** | **89.26** | **88.20** |

plausible bridging hypotheses, patch masking improves robustness to local variations, and image masking promotes coherent shape completion under diverse missing patterns, together providing complementary guidance for robust and generalizable hypothesis generation.

| Method | False Merge | | | False Split | | | False Merge&Split | | | All | | |
|---|---|---|---|---|---|---|---|---|---|---|---|---|
| | Pre/% | Rec/% | F1/% | Pre/% | Rec/% | F1/% | Pre/% | Rec/% | F1/% | Pre/% | Rec/% | F1/% |
| *Heuristic* | | | | | | | | | | | | |
| Random | 89.59 | 92.73 | 91.13 | 74.20 | 76.80 | 75.48 | 65.27 | 63.93 | 64.59 | 84.16 | 86.45 | 85.29 |
| Larger | 90.58 | 92.56 | 91.56 | **78.75** | **80.58** | **79.65** | 68.75 | 65.88 | 67.30 | 86.01 | 87.14 | 86.57 |
| Smaller | 86.56 | 92.08 | 89.24 | 70.41 | 74.17 | 72.25 | 62.56 | 62.30 | 62.42 | 81.10 | 85.42 | 83.20 |
| *Learning-based* | | | | | | | | | | | | |
| IoU Score | 90.91 | 94.37 | 92.60 | 75.92 | 78.35 | 77.11 | 66.70 | 66.37 | 66.54 | 85.54 | 88.21 | 86.85 |
| Ours | **91.73** | **94.82** | **93.25** | 78.67 | 80.58 | 79.62 | **71.29** | **69.40** | **70.33** | **87.17** | **89.26** | **88.20** |

**Hypothesis Quality for NMS**. We evaluate alternative definitions of NMS confidence by replacing our quality hypothesis with different scoring schemes. Heuristic strategies assume quality correlates with pixel count, retaining nuclei at random, the largest, or the smallest. As a stronger baseline, IoU regression predicts the true IoU from pooled 3D U-Net features with an MLP. Results in Tab. 4 show that random or small-nuclei retention performs poorly, while preferring larger nuclei only helps in false splits. In contrast, our Ordinal Quality Estimator (OQE) provides a learned hypothesis of relative completion quality, enabling more consistent confidence estimation. OQE achieves the best overall performance, improving F1 by 1.35% over IoU regression, confirming that ranking-based hypothesis quality yields a more robust NMS confidence than absolute IoU prediction.

**Feature Map Used For Hypothesis Quality Prediction**. We analyze the feature maps used for hypothesis quality prediction. Typically, the encoder's final feature map $\mathbf{x}_e$ provides strong semantic information, while the decoder's final feature map $\mathbf{x}_d$ offers richer spatial details. We evaluate $\mathbf{x}_e$, $\mathbf{x_d}$, and their combination. Specifically, we apply global average pooling to each, concatenate the vectors, and feed them into an MLP to predict $\mathcal{L}_R$. As shown in Tab. 5, the decoder feature $\mathbf{x}_d$

Table 5: Ablation studies on feature maps used for hypothesis quality prediction. Using the decoder's final feature map $\mathbf{x}_d$ achieves the best performance.

| Feature Map | False Merge | | | False Split | | | False Merge&Split | | | All | | |
|---|---|---|---|---|---|---|---|---|---|---|---|---|
| | Pre/% | Rec/% | F1/% | Pre/% | Rec/% | F1/% | Pre/% | Rec/% | F1/% | Pre/% | Rec/% | F1/% |
| $\mathbf{x}_e$ | 90.96 | 94.06 | 92.49 | 78.08 | 79.90 | 78.98 | 70.20 | 68.62 | 69.40 | 86.38 | 88.50 | 87.43 |
| $\mathbf{x}_e \oplus \mathbf{x}_d$ | 90.76 | 94.23 | 92.46 | 77.79 | 79.90 | 78.83 | 69.28 | 68.33 | 68.80 | 86.05 | 88.59 | 87.30 |
| $\mathbf{x}_d$ | **91.73** | **94.82** | **93.25** | **78.67** | **80.58** | **79.62** | **71.29** | **69.40** | **70.33** | **87.17** | **89.26** | **88.20** |

performs best, surpassing both $\mathbf{x}_e$ and the fused $\mathbf{x}_e \oplus \mathbf{x}_d$, highlighting the importance of fine-grained spatial details for accurate nucleus completion and segmentation.

## 5.4 ADDITIONAL RESULTS ON PUBLIC LIGHT MICROSCOPY IMAGES

To evaluate TANGO's generalization beyond EM, we tested on the public *C. elegans* L1 *confocal* dataset Long et al. (2009), consisting of 3D fluorescence microscopy images with dense nucleus annotations. This dataset represents a distinct imaging modality with substantially different noise and resolution characteristics from EM. As

Table 6: Results on the *C. elegans* L1 *confocal* volumes.

| Method | Train data | Pre (%) | Rec (%) | F1 (%) |
|---|---|---|---|---|
| Initial GT | N/A | 82.02 | 84.56 | 83.27 |
| Weigert et al. (2020) | Original | 80.16 | 62.69 | 70.35 |
| **Ours** | Synthesis | **86.60** | **84.84** | **85.71** |

shown in Fig. 1, the annotation of the dataset still has many obvious errors. We manually corrected all noisy segments in the released annotations using Napari (Chiu et al., 2022), which then serve as the Initial Ground Truth (GT) input to our method. Because the nuclei dataset exhibits no misalignment, we did not perform any re-alignment. For training data synthesis, we used the original noisy labels and applied augmentations such as morphological erosion and random dropout. We follow the standard train/validation split and the evaluation protocol of Weigert et al. (2020). As reported in Tab. 6, TANGO improves over the Initial GT baseline and outperforms StarDist3D, achieving +4.58% precision and +2.44% F1-score, demonstrating strong generalization across imaging modalities and biological contexts.

## 6 DISCUSSION

We presented TANGO, a tokenized *analysis-by-synthesis* framework for correcting 3D nuclei segmentations at scale. TANGO replaces fragment matching with token completion trained on synthetic data from *complete* nuclei only, and resolves overlaps via an *ordinal* selector with simple NMS. To measure progress on the last mile, we curated **NucEMFix**, a brain-wide benchmark of nuclei *error cases* across FAFB and MICrONS dataset. Across EM as well as public confocal datasets, TANGO consistently improves the three metrics over strong baselines without prompt engineering or error-specific supervision. By casting correction as completion, TANGO delivers reliable, topology-consistent fixes that better preserve nuclei integrity and reduce proofreading load. NucEMFix provides a realistic substrate for standardized evaluation and ablations, enabling reproducible comparisons and practical deployment in connectomics pipelines.

**Limitations and future work**. Tokenization currently relies on DT–watershed; learned/topology-aware tokenizers may improve splits in dense tissue. While ordinal selection is shift-robust, adding calibrated abstention (e.g., conformal risk control) and temporal consistency could further reduce over-corrections. Slice-Patch Masking models common EM/LM artifacts; incorporating PSF/scanner-conditioned priors may boost cross-modality generalization. Finally, integrating TANGO into interactive proofreading tools and quantify time savings in controlled studies.

**Reproducibility Statement**. We release NucEMFix, code, and evaluation scripts, including training configs, seeds, and data splits, to facilitate independent verification and fair comparison.

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

## A  APPENDIX

### A.1  LLM USAGE

We used LLM solely for language editing (e.g., grammar correction and phrasing improvements).

### A.2  IMPLEMENTATION DETAILS

All models trained on the NucEMFix dataset for 50 epochs using 8 NVIDIA A800 GPUs. We adopt the Adam optimizer with an initial learning rate of 0.001 and a cosine decay schedule. During the first 5 epochs, only the completion loss is optimized, after which both the completion loss and the ranking loss are jointly employed for training.

For error cases, FAFB and MICrONS volumes were padded to $128 \times 300 \times 600$ and $64 \times 300 \times 600$, respectively. The 3D U-Net completion network was trained on crops of $128 \times 128 \times 128$ for FAFB and $64 \times 64 \times 64$ for MICrONS, using 29,237 nuclei for training and 763 for validation in FAFB, and 9,000 nuclei for training and 1,000 for validation in MICrONS. SAM-Med3D is finetuned on the same data as our model trained on. Our method and SAM-Med3D directly take error cases as input without requiring additional candidates, whereas the four pair-matching baselines take each error case together with its six nearest neighbors as candidate input. For the two learning-based baselines, we trained on 15,000 synthesis cases from each dataset. False split samples were randomly generated by selecting pairs of nuclei and applying up to seven axial cuts along the $z$ axis, producing 15,000 positive pairs from the same nucleus and 15,000 negative pairs from different nuclei.

### A.3  DOES OUR PROPOSED METHOD REQUIRE ERROR DETECTION

Table A1: Results on the correctly annotated nuclei

| Method | Setting | FAFB | | | MICrONS | | |
|---|---|---|---|---|---|---|---|
| | | Pre/% | Rec/% | F1/% | Pre/% | Rec/% | F1/% |
| **Ours** | Self-Supervised | **99.99** | **99.99** | **99.99** | **99.96** | **99.96** | **99.96** |

In large-scale whole-brain error correction, most methods first detect errors and correct only the wrong annotations to avoid introducing new mistakes. Our method does not need error detection and can be applied directly to all annotated nuclei. We train the model using randomly sampled mixed data that includes both correct and incorrect annotations. Because errors are rare, they have little effect on training, so the model learns to complete structures without harming the correct ones. Before training, we simply remove samples whose bounding boxes exceed normal size thresholds to exclude obvious outliers. For training, we randomly selected 29,000 samples from FAFB and 9,000 from MICrONS. For testing, we used 30,000 and 10,000 correctly annotated nuclei from the remaining parts of each dataset. This test set ensures that we can check whether our method creates new errors when applied to accurate data. The result is shown in Tab. A1. In the FAFB test set of 30,000 correctly annotated nuclei, only one nucleus was incorrectly split into two segments. In MICrONS, only 4 of 10,000 correct nuclei were erroneously modified. This corresponds to an error introduction rate of less than 0.05%, demonstrating that our method preserves the integrity of accurate annotations and rarely introduces new mistakes.

### A.4  MORE ABLATION STUDIES

**Realignment Pre-processing**. To evaluate the necessity of the realignment pre-processing step, we conducted experiments with and without this operation. The NucEMFix-F dataset contains 329 misaligned samples, which lead to thousands of error nuclei segmentations. As shown in Tab. A2, evaluating these samples specifically, the F1 score increases from 41.11% to 73.46% with realignment, highlighting its crucial role in correcting alignment-related errors. The improvement is especially notable for the false split and false merge&split samples. However, these results also indicate that misalignment remains a challenging issue, leaving room for further gains in robustness and generalization.

Table A2: Ablation studies on realignment pre-processing. For each metric, the first number shows the result on all samples, and the second number shows the result on the subset of misaligned samples.

| Error Type / Metric | w/o realign | with realign |
|---|---|---|
| **False Merge** | | |
| Pre / % | 94.62 / 67.21 | 91.73 / 71.28 |
| Rec / % | 95.77 / 71.26 | 94.82 / 81.32 |
| F1 / % | 95.19 / 69.18 | 93.25 / 75.97 |
| **False Split** | | |
| Pre / % | 77.60 / 20.63 | 78.67 / 69.70 |
| Rec / % | 77.38 / 20.00 | 80.58 / 70.77 |
| F1 / % | 77.49 / 20.31 | 79.62 / 70.23 |
| **False Merge&Split** | | |
| Pre / % | 39.83 / 36.15 | 71.29 / 73.74 |
| Rec / % | 32.55 / 29.30 | 69.40 / 72.09 |
| F1 / % | 35.83 / 32.37 | 70.33 / 72.90 |
| **All** | | |
| Pre / % | 85.81 / 43.74 | 87.17 / 72.70 |
| Rec / % | 84.33 / 38.78 | 89.26 / 74.24 |
| F1 / % | 85.06 / 41.11 | 88.20 / 73.46 |

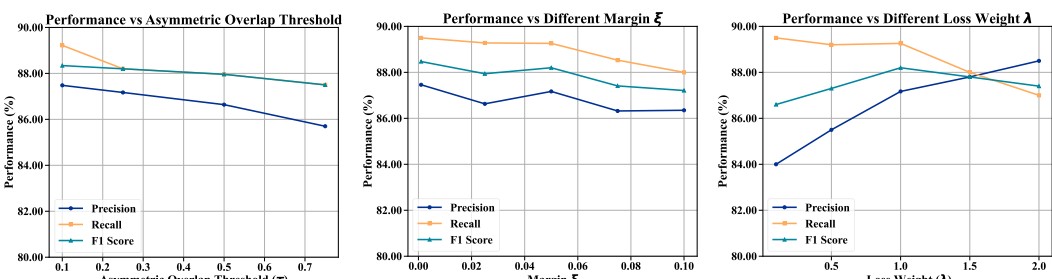

Figure A1: Ablation studies on hyperparameter $\tau$ and iteration of erosion.

**Hyperparameter analysis**. Fig. A1 presents our hyperparameter analysis, examining the overlap threshold $\tau$ in NMS, the margin $\xi$ in $\mathcal{L}_R$, and the weight $\lambda$ for $\mathcal{L}_R$. As the overlap threshold $\tau$ increases from 0.1 to 0.75, F1 decreases only slightly from 88.3 percent to 87.5 percent, indicating that the model is robust to $\tau$. Similarly, varying the margin $\xi$ in $\mathcal{L}_R$ has minimal effect on F1, which fluctuates between 87.2 percent and 87.9 percent. Setting the loss weight $\lambda$ to 1.0 yields the best F1. Overall, our method is not sensitive to hyperparameter choices.

## A.5 FAILURE CASES

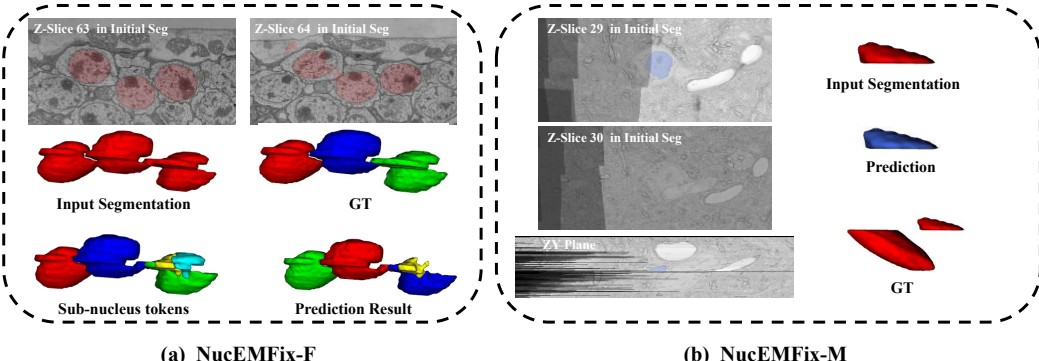

(a) **NucEMFix-F**  (b) **NucEMFix-M**

Figure A2: Failure cases on NucEMFix.

TANGO failures are mainly caused by defective slices. In Fig. A2 (a), slices 63 and 64 exhibit non-strict translational misalignment due to slice distortion, and the annotations on these defective slices are unreliable. As a result, TANGO cannot recover complete shapes for each token, and our realignment pre-processing also fails. In Fig. A2 (b), the false split cases of MICroNS predominantly occur near the boundary of the volume. Illumination inconsistencies similarly impair realignment, so our predictions show no improvement compared to the original false splits.

## A.6 MORE STATISTICS OF NUCEMFIX

Fig. A3 presents a detailed statistical analysis of error cases in NucEMFix and FAFB and MICrONS data are mixed together for presentation. The left and right figures show the distribution of the number of nuclei covered by each error sample and the distribution of the voxel count per error sample in the NucEMFix dataset, respectively.

It can be observed that the distribution of error samples exhibits significant variability and imbalance. The majority of error samples involve 2-10 nuclei. However, a notable number of samples encompass 30-64 nuclei; in fact, these are mostly false merge&split samples with misalignment, representing more challenging samples. The voxel distribution is similar to the nuclei count distribution. False merge or false split samples that involve more nuclei tend to produce larger error samples.

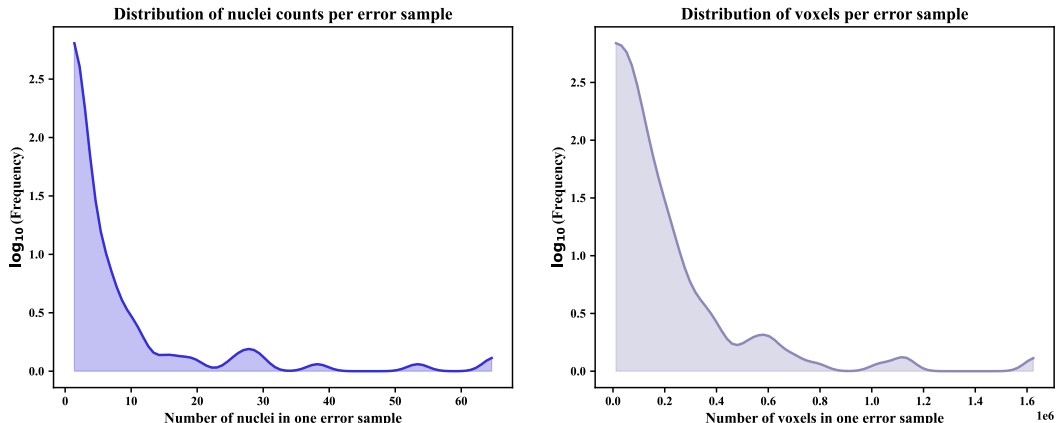

Figure A3: Distribution of nuclei counts and voxels per error sample in NucEMFix.

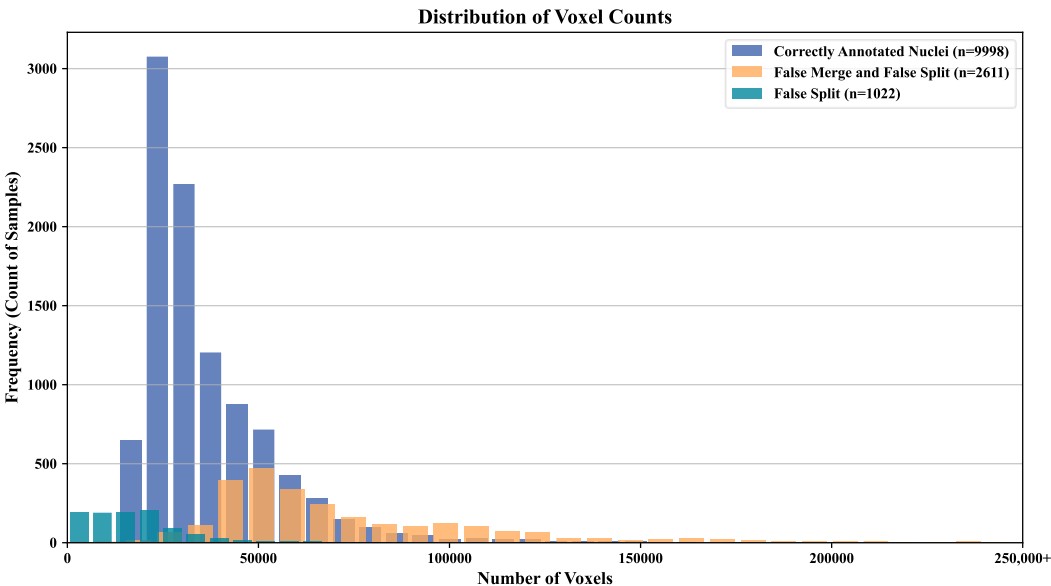

Figure A4: Distribution of voxels per sample in NucEMFix-F.

Fig. A4 shows the distribution of voxel counts per sample in NucEMFix-F. We randomly sampled approximately 10,000 nuclei from the correctly annotated nuclei. While false split errors tend to have fewer voxels and false merge&split or false merge errors tend to have more, there is substantial overlap among all groups. This indicates that voxel count alone is insufficient for reliably distinguishing erroneous samples from correct annotations.

## A.7 DETAILS OF REALIGNMENT PRE-PROCESSING

To address misalignment along the $z$-axis, we first identify bad slices as those with more than 30% of black voxels or an intensity variance below 0.1, and treat them as missing. For each valid slice, we compute its relative displacement to the nearest non-bad neighbor using patch-based phase correlation (Li et al., 2019). These displacements are accumulated across the volume to achieve global realignment. The segmentation masks on bad slices removed to prevent error propagation.

---

**Algorithm 1** Patch-based Phase Correlation

---

**Input:** Volumetric image stack $V \in \mathbb{R}^{Z \times H \times W}$
**Output:** Realigned image stack $V_{\text{aligned}}$, displacement map $D \in \mathbb{R}^{Z \times 2}$

 1: Initialize displacement map: $D \leftarrow \mathbf{0}^{Z \times 2}$
 2: Identify bad slices: $\mathcal{B} \leftarrow \{z \mid \text{black voxels or low intensity}\}$
 3: **for all** slice $z \notin \mathcal{B}$ **do**
 4:     Find nearest valid neighbor slice $z_{\text{neighbor}}$
 5:     Extract non-overlapping patches $I^{(i)}$ from slice $z$ and $J^{(j)}$ from $z_{\text{neighbor}}$
 6:     **for all** patch pair $(I^{(i)}, J^{(j)})$ **do**
 7:         Normalize each patch to zero mean
 8:         Compute FFT: $F_I^{(i)} \leftarrow \mathcal{F}[I^{(i)}]$, $F_J^{(j)} \leftarrow \mathcal{F}[J^{(j)}]$
 9:         Cross-correlation:
$$C^{(i,j)} \leftarrow \mathcal{F}^{-1}\left(F_I^{(i)} \cdot \overline{F_J^{(j)}}\right)$$
10:         Peak location:
$$(x^{(i,j)}, y^{(i,j)}) \leftarrow \arg\max \text{fftshift}(\Re(C^{(i,j)}))$$
11:         Displacement estimate:
$$\Delta x^{(i,j)} \leftarrow x^{(i,j)} - \lfloor W/2 \rfloor, \quad \Delta y^{(i,j)} \leftarrow y^{(i,j)} - \lfloor H/2 \rfloor$$
12:         **if** $|\Delta x^{(i,j)}| > T$ **or** $|\Delta y^{(i,j)}| > T$ **then**
13:             **continue**                                      ▷ Discard this patch pair
14:         **end if**
15:         Quality score:
$$q^{(i,j)} \leftarrow \frac{\max C^{(i,j)}}{\text{mean}(C^{(i,j)}) + \delta}$$
16:     **end for**
17:     Fuse remaining displacements using weighted median:
$$D_z^x \leftarrow \text{WMedian}\left(\{\Delta x^{(i,j)}\}, \{q^{(i,j)}\}\right), \quad D_z^y \leftarrow \text{WMedian}\left(\{\Delta y^{(i,j)}\}, \{q^{(i,j)}\}\right)$$
18: **end for**
19: Apply accumulated displacements $D$ to realign image stack $V$
20: **return** $V_{\text{aligned}}, D$

---

The details of the patch-based phase correlation used in realignment pre-processing is shown in Alg. 1. We denote the displacement map by $D \in \mathbb{R}^{Z \times 2}$, where each entry $D_z = (D_z^x, D_z^y)$ represents the in-plane shift of slice $z$ with respect to its nearest valid neighbor. Each slice is first normalized to zero mean and unit variance, and then divided into overlapping patches of size $256 \times 256$ using a sliding window with stride 32. A patch is considered invalid if it has very low intensity variation or more than 30 percent of its pixels have intensity below 80. Bad slices are identified as those for which a significant fraction of patches are invalid. Their relative displacement with respect to the nearest valid slice is set to zero, and they are aligned directly by inheriting the cumulative shift of that reference slice. For the remaining slices, phase correlation is computed only between valid patch pairs $I^{(i)}$ and $J^{(j)}$. The inverse Fourier transform of the cross-power spectrum of each patch pair yields a correlation response $C^{(i,j)}$, whose peak location provides a patch-wise displacement estimate $\Delta^{(i,j)} = (\Delta x^{(i,j)}, \Delta y^{(i,j)})$. Patch pairs whose displacement magnitude exceeds 50 pixels

in either direction are discarded to prevent unreliable matches. To evaluate the reliability of each remaining estimate, we assign a quality score $q^{(i,j)}$ defined as the ratio between the peak value and the mean correlation response. Patch displacements are capped at a maximum of 128 voxels in each direction to prevent excessive shifts. All patch displacements are then fused by computing a quality-weighted median, which ensures robustness against spurious matches and local image degradations. The resulting displacement map $D$ is subsequently accumulated and applied to the image stack to produce the realigned volume.

