# OpenReview forum: "TANGO: Tokenized Analysis-by-synthesis for 3D Nuclei Segmentation Correction"
_ICLR.cc/2026/Conference — ICLR 2026 Conference Withdrawn Submission_

### Official Review · Reviewer_pM5H · 2025-10-28

**Soundness:** 3
**Presentation:** 3
**Contribution:** 3
**Rating:** 6
**Confidence:** 3

**Summary:**

This work proposes TANGO, a tokenized analysis-by-synthesis framework that does 3D NUCLEI SEGMENTATION CORRECTION, and NucEMFix, a brain-wide benchmark of nuclei error cases in large-scale EM volumes (FAFB, MICrONS) with diverse, realistic failures to drive research on last-mile correction. The new benchmark contains more than 8k 3D error-affected nuclei,

**Strengths:**

1. Data cleaning is always an important step, and the automatic annotation correction approach has the potential to speed up the manual annotation procedure.

2. The tokenized analysis-by-synthesis framework is well-designed. It breaks down erroneous masks into smaller nucleus fragments and generates several completion hypotheses, which eliminates the need for paired error labels.

3. NucEMFix provides a major benefit to the community. The dataset includes over 8,000 systematically annotated error cases across two major electron microscopy datasets (FAFB and MICrONS) with around 250 hours of expert annotation work. It describes different types of errors (false merges, false splits, and false merge & splits) and failure modes (misalignment, missing slices, and close-set contacts). This fills a significant gap in standardized evaluation for "last-mile" correction issues. Including both EM and confocal data shows its use in multiple contexts.

4. Experimental results showed significant improvements by TANGO. The comprehensive ablation studies of training data synthesis strategies, the design of the ordinal quality estimator, and the selection of feature maps explain the reason for the performance boost.

**Weaknesses:**

1. Computational costs were not discussed. How long does it take to process the entire volumes? What GPU memory is required for the 128 stacks? The tokenization step involves watershed on every instance, completion network inference for each token, and non-maximum suppression. These processes could be too costly at the petabyte scale mentioned in the abstract.

**Questions:**

None

---

### Official Review · Reviewer_zgrc · 2025-11-01

**Soundness:** 2
**Presentation:** 3
**Contribution:** 2
**Rating:** 2
**Confidence:** 4

**Summary:**

The paper proposes a method ("TANGO") to correct the "last-mile" errors in nuclear segmentations. The approach is based on heuristic splitting of existing nuclei, prediction of multiple extension hypotheses of the split fragments with a U-net network, followed by a final selection procedure based on a trained quality estimator. The authors show improved segmentation results on two large public EM volumes, as well as on an additional confocal volume.

**Strengths:**

- New dataset of 8k+ corrected nuclei (NucEMFix) from FAFB and MICRONS.
- TANGO model training requires positive labels only.
- The paper includes evaluation on data from another domain (light microscopy)
- Multiple ablations included in the evaluation.

**Weaknesses:**

- The paper calls splitting existing segments into smaller ones with a watershed transform "tokenization". I find this confusing and not aligned with common terminology used in the computer vision community. "Subsegments", "supervoxels" etc. would be more natural terms here.

- Incompletely acknowledged prior work:

1) The on-the-fly realignment is not novel, and has been applied to one of the datasets used in the present paper (FAFB) previously in Li et al., 2019 in the context of general instance segmentation. That paper is cited in the appendix, but this prior art should also be acknowledged in the main text. It should also be noted that in newer datasets severe misalignments of this kind are very rare or completely eliminated thanks to progress in alignment tools and acquisition techniques.

2) Slice masking during training is also a well established procedure in EM data processing, and the authors should cite relevant early work, e.g. https://arxiv.org/abs/1706.00120.

3) The concepts of "analysis by synthesis" and "object completion" have been in used in flood-filling network (FFNs) agglomeration and segmentation, respectively (https://doi.org/10.1038/s41592-018-0049-4) in the context of general EM segmentation. Like TANGO, these models also use positive labels only.

- As noted in the text, the simple watershed-based splitting accounts for most of the performance gains for "false merge" case, thus leaving "false splits" as main category in which improvements are needed. For a clean comparison, the baselines should include standard EM segmentation+agglomeration methods like affinity prediction or flood-filling networks, trained on the same nuclei-specific data as TANGO. This would help elucidate the impact of the primary novelty of the present approach, which, given the prior work listed above, is in the use of an "Ordinal Quality Estimator" + NMS.

**Questions:**

- line 295: it's a bit confusing to specify resolution in zyx order; xyz (in-plane, axial) is far more typical.
- line 341: what is SCNF?
- line 456: which "Fig. 1" is this referring to?
- Why is Fig. 5 not referenced in the text and why does it only include results for the -F subset?
- In Fig. 5, in the "false merge" case, what explains very similar performance of four very different baseline methods?

---

### Official Review · Reviewer_48HC · 2025-11-01

**Soundness:** 2
**Presentation:** 3
**Contribution:** 2
**Rating:** 2
**Confidence:** 4

**Summary:**

This paper introduces TANGO a "tokenized analysis-by-synthesis" framework for correcting segmentation errors in 3D microscopy. Given a problematic seed mask, TANGO decomposes it to tokens, generates multiple hypotheses using a U-Net trained on only complete nuclei, and ranks the results with an ordinal quality estimator followed by non-maxima suppression.

The authors introduce a new benchmakr NucEMFix containing 8k manually curated nuceli errors. The method achieves good F1 scores and shows some cross-modality generalization to confocal data.

Overall while the method appears sound, the limited technical novelty and limited scope make this unsuitable for ICLR in its current form.

**Strengths:**

* Addresses a clear practical bottleneck in connectomics
* The pipeline is clearly described and reasonable, with appropriate ablations
* The contribution of the dataset is valualble to the community

**Weaknesses:**

* Limited conceptual novelty - the method reuses standard components such as watershed tokenization, U-Net, ranking based slection with no new learning principle. The analysis-by-synthesis framing is questionable as there is no generative loop or analysis of learned priors
* evaluation are all processed through TANGOs tokenization which favors it as a solution
* narrow scope - the impact is and interest is limited in terms of the ICLR community

**Questions:**

1. Can you report baselines without TANGO tokenization to assess genuine improvement?

---

### Official Review · Reviewer_R8QH · 2025-11-01

**Soundness:** 2
**Presentation:** 1
**Contribution:** 3
**Rating:** 2
**Confidence:** 4

**Summary:**

The paper proposes TANGO, a framework for 3D nuclei segmentation correction. Specifically, tehy focus on the *last mile* error problem, caused by rare, heterogeneous false merges/splits and missing-slice or misalignment artifacts. They aim to close the gap and have accurate 3D nuclei segmentation for very large images of mouse brain tissue and brain Drosophila from electron microscopy. Their framework uses a tokenized analysis-by-synthesis for 3D nuclei segmentation correction. The 3D UNet is trained on masked complete nuclei to generate completion hypothesis, which are then selected by a lightweight ordinal selector. They evaluate against a few simple baselines, MED-SAM, and against a public C. elegans dataset.

They also curate the NucEMFix dataset, which took over 200 hours of annotator efforts to correct the nuclei error cases across the FAFB and MICrONS datasets.

**Strengths:**

- **Problem**: The authors tackle a meaningful problem in nuclei segmentation: correcting the long tail of errors needed to obtain accurate 3D segmentations of petabyte scale images.

- **Formulation**: A key contribution of this work is to propose a generative process to complete the multiple shape hypotheses rather than a discriminative approach. They also can train without needing error labels using slice-patch masking.

-  **Dataset**: The NucEMFix dataset is a valuable contribution to the community and took significant man hours to create.

- **Experimental Performance**: The authors demonstrate strong experimental performance improvements over simple baselines and Med-sAM3D. They also outperform the StarDist3D benchmark on the public C. elegans dataset. Several ablations were also performed in the appendix.

**Weaknesses:**

While this paper tackles an important issues and presents impressive results, there are several weaknesses that make it not ready for publication.

**Clarity of writing**: The paper is very difficult to follow. The motivation for the tokenized analysis by synthesis is vague. The method description is somewhat disconnected, and the diagram does not do a good job of describing what is going on. I have a hard time following how the different components fit together. There is a lot of implementation detail in the body of the text that distracts from the main message, and could be moved to the appendix. The paper is also notation-heavy and lacks intuitive figures. It requires multiple passes to understand the pipeline.

**Limited technical novelty**: The core approach describes a standard patch-based completion model with a scoring framework. I appreciate the use of synthetic data in training, but even the corruption model is too simple. As I understand, the model is essentially just a masking to the nuclei, predicting segmentations with a U-Net, and performing ranking. The term analysis by synthesis seems to be doing a lot of heavy lifting for a simple model. Is the slice-patch masking even sufficient enough of a corruption to learn on? Can the authors comment on using more sophisticated synthesis techniques, such as diffusion or GMMs? Further, the method seems to be overly engineered and heuristic heavy.

**Weak related work discussion**: The paper does not do a good job of outlining the prior related work in this area. Really, the only model that is mentioned is Med-SAM3D, and that is not fair to other work. For example, StarDist3D and AnyStar are both models that have been developed explicitly for nuclei segmentation. AnyStar (Dey et al. 2024), actually proposes a fully synthetic training framework for this task. A more comprehensive related work should focus on additional instance segmentation correction models, and 3D generative models. Additional semi-automatic segmentation models should be included, such as MultiVerSeg (Wong et al., ICCV 2025).

**Insufficient baselines**: Building on the previous point, the comparison set is thin. The author proposes a few very simple heuristic models that would not actually be used in practice, and Med-SAM3D, which was not trained for this task. The authors should use modern baselines on 3D nuclei segmentation (for example, AnyStar) or shape-completion works, or foundation models for microscopy.

**Evaluation**: I have a hard time understanding the evaluation and splitting of training and testing data. As I understand, the authors used the entire images for training, and labels were created synthetically. The evaluation was then done on the corrected labels from the NucEMFix dataset. Was the NucEMFix dataset used in training? I have a hard time ruling out the model being trained on the author-curated data simply degraded from synthetic corruptions. There are no cross-dataset experiments demonstrating genuine generalization.

The authors also reported needing 8 A800 GPUs to train this model. Why did it require so much computational power? Would practitioners actually be able to use this model?

**Questions:**

Can the authors provide a high-level motivation for why tokenization improves?

Can the authors comment on the use of Vision Transformers, which naturally take in tokens?

Why were there no evaluations with standard nuclei segmentation models or foundation models?

What can these results enable in neuroscientific research?

---

### Note · Authors · 2025-11-15

I have read and agree with the venue's withdrawal policy on behalf of myself and my co-authors.